# Lipid Differences and Related Metabolism Present on the Hand Skin Surface of Different-Aged Asiatic Females—An Untargeted Metabolomics Study

**DOI:** 10.3390/metabo13040553

**Published:** 2023-04-13

**Authors:** Tian Chen, Juan Wang, Zhenxing Mao

**Affiliations:** 1Division of Public Health Service and Safety Assessment, Shanghai Municipal Center for Disease Control and Prevention, Shanghai 200336, China; 2NMPA Key Laboratory for Monitoring and Evaluation of Cosmetics, Shanghai 200336, China; 3College of Public Health, Zhengzhou University, Zhengzhou 450001, China

**Keywords:** skin surface lipids, lipidomic analysis, female hand

## Abstract

This cross-sectional study aimed to investigate differences in skin surface lipids (SSL) and explore related metabolic pathways among females of different ages in Henan Province. Ultra-performance liquid chromatography quadrupole time-of-flight mass spectrometry (UPLC-QTOF-MS) was used to determine the lipid composition of the skin surface of 58 female volunteers who were divided into three age groups. Statistical analysis was performed using Progenesis QI, Ezinfo, and MetaboAnalyst. Multivariate and enrichment analysis were used to identify the different SSL among the groups. A total of 530 lipid entities were identified and classified into eight classes. Among these, 63 lipids were significantly different between the groups. Lower levels of glycerolipids (GLs) and sphingolipids (SPs) were observed in the middle-aged group, while higher levels of GLs were found in the elder group. GLs belonged to the largest and statistically significant enrichment of lipid metabolic pathways, and the lipid individuals enriched to the sphingoid bases metabolism were the most and statistically significant. These findings suggest that there are differences in hand SSL among females of different ages, which may be related to GLs and sphingoid bases metabolism.

## 1. Introduction

The skin is the largest and outermost organ of the human body and has a variety of defense and regulatory functions [1]. The stratum corneum (SC), formed by terminally differentiated keratinocytes in the outermost layer of the epidermis, mainly provides the skin surface barrier function [2]. The skin surface barrier is composed of a “brick–mortar” structure, where the “bricks” are non-nucleated stratum corneum cells, and the “mortar” is a lipid-rich extracellular matrix [3,4]. The lipids on the skin surface, composed of ceramides, cholesterol, free fatty acids (FAs), and some small molecular lipids, play a crucial role in maintaining the skin barrier function [5]. These lipids are derived from precursor cells and are then “lipid processed” by enzymes released from the epidermal lamellar bodies into intercellular lipids [6]. Any factor that affects the skin surface acidity, Ca+ concentration gradient, and barrier function can impact lipid turnover [7].

The natural aging process of the skin leads to a loss of elasticity, dryness, fine lines, atrophy, and laxity of the skin. The two main mechanisms responsible for these changes are intrinsic aging and environment-dependent extrinsic aging, caused by exposure to sunlight-related UV radiation (photoaging), smoking, and air pollution, among others [8,9]. The stratum corneum’s barrier function is also incapable of resisting the aging process, and age-related skin changes can lead to visible signs of aging and structural and functional impairments [10]. A damaged epidermal permeability barrier (EPB) leads to transepidermal water loss (TEWL) and triggers inflammatory skin diseases that negatively impact the quality of life [11]. Many skin diseases, such as lamellar ichthyosis, psoriasis, Netherton syndrome (NTS), and atopic dermatitis (AD), are characterized by defective or weakened epidermal barrier function [12]. In addition, the prevention of skin aging and skin rejuvenation are becoming areas of active research. Aging is inherently complex, and it is difficult to quantify and interpret experimental results. Previous studies on changes in barrier function with age have been controversial. Therefore, new technology is needed to explore the damage mechanism of skin barrier function with age. This will enable us to quantify the degree of damage and provide a theoretical basis for the development and optimization of corresponding skin care products.

Lipidomics is a powerful analytical tool that utilizes liquid chromatography–mass spectrometry (LC–MS) machines to study human lipid metabolites through the principles of analytical chemistry [13]. It is widely used in various fields, including cancer [14], metabolic syndrome [15], and skin diseases [16] due to its ability to evaluate all lipids in biological systems and conduct large-scale lipid research [13]. By analyzing and identifying lipid differences between different populations, lipidomics can help explore differential pathways of lipid metabolism, which provides a basis for studying the pathogenesis of diseases. Previous studies have investigated the characteristics of skin aging by distinguishing between parts of the skin that are subjected or not to sunlight. In particular, facial skin has been actively studied because it is not only the part that is exposed to the outside world, but it is also the part of the body that attracts more aesthetic attention than other parts of the body [17]. Conversely, limited research has been conducted on hand skin, despite the fact that it is also exposed and is the most vulnerable structure besides the head to environmental stresses (abrasions, cuts, lacerations, and chemical and thermal burns). In addition, unlike the skin on the surface of the hand and palm, the skin on the back of the hand has relatively abundant sebaceous glands [18]. The skin on the back of the hand was selected as a sampling point for this study. Meanwhile, gender differences in skin have to be considered [19]. Compared with males, females in rural China are generally more likely to engage in domestic work and to be exposed to detergents or solid particles associated with cooking [20,21]. Therefore, a cross-sectional study was conducted, which recruited a group of healthy female subjects of different ages. Non-invasive skin detection technology and lipidomics were used to evaluate the skin barrier function of people of different ages and explore the related metabolic pathways.

## 2. Materials and Methods

### 2.1. Chemicals and Reagents

LC–MS-grade acetonitrile (ACN, catalog no. A955-4), methanol (MT, catalog no. A456-4), ammonium formate (catalog no. A115-50), isopropyl alcohol (IPA, catalog no. A461-4), N-propanol (NPA, catalog no. A414-4), acetonitrile (catalog no. A955-4), and formic acid (catalog no. LS118-4) were obtained from Thermo Fisher (Waltham, MA, USA). skin surface lipids (SSL)-adsorbent tapes Corneofix^®^ (Cologne, Germany) and a Mass Spectrometer (Waters UPLC-VION QTOF MS, Milford, MA, USA) were used for this study.

### 2.2. Study Participants

Female volunteers aged 18 to 79 years from rural Henan, China, were recruited for the study. The inclusion criteria were as follows: (a) healthy volunteers aged 18–79 years, who were divided into a younger group: 18–44 years old, middle-aged group: 45–59 years old, and elder group: 60–79 years old; (b) no previous common skin diseases, such as contact dermatitis, eczema, urticaria, etc.; (c) refrain from using any skincare or washing products for at least 3 days before testing; (d) do not smoke, drink, or stay up late. (e) do not spend more than 4 h outdoors every day. The exclusion criteria included: (a) not persisting until the end of the experiment; (b) sensitive skin. Due to age differences in the biophysical characteristics of skin, in this study, according to the guidance of WHO and the actual situation in our country, the subjects were divided into three age groups to analyze the changes in SSL [22]. Finally, a total of 58 subjects were enrolled in the study. All participants were informed about the study’s purpose and methods and signed a written informed consent form. The study was conducted with the approval of the Ethics Committee of Zhengzhou University and conforms to the principles of the Declaration of Helsinki.

### 2.3. Sample Collection and Preparation

The collection of skin lipid samples was performed in May 2020. Before collecting samples, the righthand surface was cleaned with clean water and dried with a non-woven fabric. After sitting indoors for 30 min (the temperature was kept at 20 ± 1 °C, and the relative humidity was about 40–60 %), the experimenter attached the Corneofix^®^ test strips to the back of the volunteer’s right hand. The strips were removed after 30 min, placed in 2 mL EP tubes, and stored frozen at −80 °C.

For preparation, the samples were retrieved from the freezer, and 1.5 mL of methanol was added; then, they were vortexed for 60 min and allowed to settle at −20 °C for 12 h. The Corneofix^®^ test strips were retrieved from the EP tubes, and the remaining lipid-containing extract was blow-dried using the nitrogen evaporator (Hengao, China). Finally, the residue was redissolved in 200 µL of IPA/ACN/distilled water (65/30/5) and transferred to a vial for detection after vortexing.

### 2.4. Ultraperformance Liquid Chromatography Analysis

Chromatographic separation (Waters, USA) was performed on an ACQUITY UPLC CSH C18-column, 2.1 × 100 mm, 1.7 µm particle size. The flow rate was set at 0.3 mL/min, the column temperature was kept at 50 °C, and the sample injection volume was 5 µL. Mobile phase A was composed of a solution of 10 mM ammonium formate with 0.1% formic acid in ACN/water (60:40, *v*/*v*), while mobile phase B was composed of a solution of 0.1% formic acid in IPA/ACN (90:10, *v*/*v*). Gradient elution was performed with a mixture of mobile phase A and mobile phase B (details provided in Table 1).

### 2.5. Mass Spectrometry Analysis

Mass spectra were generated by electrospray ionization (ESI) in the positive mode, with a range of *m*/*z* 50 to 1000 for scanning. Data acquisition was performed using MSE in the continuum mode, with leucine encephalin (*m*/*z* 554.2771) as the lock mass for mass correction. In the positive ion mode, the capillary voltage was set to 3000 V, the source temperature was set to 120 °C, and the desolvation temperature was set to 500 °C. The cone gas and desolvation flow rate were 50 L/h and 900 L/h, respectively (see Appendix A).

### 2.6. Data Extraction and Analysis

First, all mass spectral data of the samples were imported into Progenesis QI 3.0.3 (Waters Corporation) for data processing, including peak alignment, peak extraction, normalization, etc. The Lipid Maps Structure Database (LMSD) (http://www.lipidmaps.org, accessed on 15 August 2022) and the human metabolome database (HMDB) (https://hmdb.ca/, accessed on 16 August 2022) were performed to identify the compound ID of the lipid composition of each sample. Second, lipid composition data were imported into MetaboAnalyst 5.0 (https://dev.metaboanalyst.ca/, accessed on 29 August 2022) to explore the distribution of SSL between the three groups by heatmaps. Third, the preprocessed data imported into QI were grouped into three groups and then imported into Ezinfo (Waters Corporation) for analysis. Principal component analysis (PCA) and supervised partial least-squares discriminant analysis (PLS-DA) were performed to visualize the metabolic and lipid differences between the groups. The corresponding loading plot was used to determine the metabolites most responsible for separation in the PLS-DA model. Based on these results, important differential metabolites were evaluated by the values of variable importance in the projection (VIP), fold change (FC) and *p*-value, with the criteria of simultaneously meeting VIP > 1, |FC| ≥ 2, and *p*-value < 0.05. In addition, significant differences between relative abundances in groups were assessed with the one-way Analysis of Variance (ANOVA) with SPSS software version 21.0, and a two-tailed *p* < 0.05 was considered statistically significant. Finally, we used metabolite set enrichment analysis (MSEA) to investigate lipid metabolism pathways that were enriched by differentiated lipids in MetaboAnalyst.

## 3. Results

### 3.1. Differences in Right-Hand SSL in Females of Different Age Groups

The number of subjects in the three groups was 20 in the younger group, 19 in the middle-aged group, and 19 in the elder group. The age distribution of the participants was presented as the median (mean) respectively: 32.5 (33.4) y; 55 (51.9) y; and 66.5 (66.8) y. The 530 lipid entities were identified in the righthand SSL in three groups. A heat map was used to show the distribution of SSL at different ages of females. As shown in Figure 1a, the top 30 substances were singled out for display according to VIP. There was a significant separation of three groups, explaining that there were differences in right-hand SSL among them. Then, all identified lipids were classified into eight different classes. The relative content of each major lipid was calculated to compare the differences in lipids in females of different ages. Compared with the younger group, Figure 1b illustrated that the sterol lipids (STs) significantly decreased and the other lipids increased, including sphingolipids (SPs), prenol lipids (PRs), polyketides (PKs), and FAs, in the middle-aged group. Moreover, SPs and glycerolipids (GLs) increased significantly in the elderly group, while STs decreased (*p* < 0.05). It is worth noting that the pattern of expression of GLs and SPs seems to indicate that the younger group expresses fewer of these lipid types than the two older groups (all *p* < 0.05).

### 3.2. Multivariate Data Analysis of Right-Hand SSL in Females of Different Age Groups

#### 3.2.1. Sixty-Three Individual Lipids Were Responsible for the Differences between Skin Surface Lipids in Skin Samples

PCA and PLS-DA models were used for multivariate data analysis. The resulting model score plot is shown in Figure 2a,b. These results demonstrated substantial separation and significant differences among the three different age groups of SSL samples (R_2_X = 0.52, R_2_Y = 0.82, Q_2_ = 0.56). According to the loading plot in Figure 2c, after the PCA of the main lipid with a condition of VIP ≥ 1, the differences among the three groups were further selected. Finally, there were 63 significantly different lipids that were finally identified in Table 2. The selection criteria were VIP ≥ 1, *p* < 0.05, and fold change >2.

#### 3.2.2. Classification and Relative Abundance Alterations of Specific Differentiated Lipid Metabolites in Skin Samples

The corresponding metabolites based on the different peaks were divided into six categories through searching the LMSD and HMDB databases, which are presented in Figure 3a. In general, GLs (49.21%), FAs (25.40%), and SPs (19.05%) constituted the majority of the 63 substances. As these classes were predominant among these lipids, we focused on these groups of compounds. In this study, a relative content comparison of these lipid species was analyzed. In Figure 3b, GLs, FAs, and SP levels were lower in the middle-aged group and higher in the elders. It was found that only lower levels of GLs and SPs were statistically significant in the middle-aged group, while higher levels of GLs were found for the elders. Similar observations were made regarding STs and PRs, which seem to be present in the younger group skin (<44 y) and nearly absent in the two older groups combined (>45 y).

The heatmap in Figure 4a shows all the differentiated lipids of the skin samples, and it can clearly be seen that there were differences between the three age groups.

### 3.3. Enrichment Analysis of Skin Lipid Metabolism of Right-Hand SSL in Females of Different Age Groups

Lipid metabolism profiles were obtained by the enrichment analysis of SSL in three different age groups of females. Figure 4b showed that the enrichment analysis module identified fourteen kinds of metabolism associated with skin conditions. Obviously, there was no statistical significance for lipid metabolism in the lighter-colored circle at the bottom. From yellow to red, the *p*-value is from large to small, indicating that the degree of enrichment is becoming more and more significant. There are lipid pathways on the vertical axis and enrichment factors on the horizontal axis; the higher the value, the greater the degree of enrichment. The size of the dots represents the number of lipids enriched in this pathway. Our results showed that the most statistically significant and largest enrichment lipid metabolism was GLs. In addition, the lipid individuals enriched to the sphingoid bases metabolism were the most and statistically significant.

## 4. Discussion

Focusing on the status of skin in different life courses, we describe the application of UPLC-QTOF-MS-based lipidomics in the study of SSL differences in different age groups. This approach provides comprehensive information on the mechanisms underlying differences in skin status across ages. In our study, we validated the differences in SSL in different age groups through lipidomic models. By searching the database, the differentiated metabolites were classified and compared according to their relative abundance to try to obtain the change in skin lipids at different ages.

Overall, we identified a total of 530 lipids and observed that, among the eight main classes of lipids, the pattern of expression of GLs and SPs seems to indicate that the younger group (<44 y) expresses fewer of these lipid types than the two older groups combined (>45 y). After statistical processing, it was found that surface skin differential lipids were mainly concentrated in FAs, GLs, and SPs. The levels of GLs, FAs, and SP among females in the middle-aged group were lower than those of younger females, while the levels were higher among elders. However, there was only a statistically significant difference between lower levels of GLs and SPs in the middle-aged group and the higher levels of GLs in the elder group. In addition, similar observations have been made regarding STs and PRs. These seem to be present in the younger group skin and nearly absent in the two older groups together. It would be interesting to identify several representative lipids that can be used as examples for these lipid families.

As the research progressed, techniques such as the analysis of stratum corneum lipid components, including tape stripping, and corneal surface measurement were being used to measure the effects of interventions on (elderly) skin and 3D organ models [23]. It has been confirmed that both intrinsic and extrinsic aging have an effect on the morphology of corneal cells, with the surface area increasing or decreasing [24]. One study using scanning electron microscopy found that two opposing aging effects resulted in no significant differences in physical aging between hand samples from two age groups [25]. Unlike internally aging skin, external aging (UV irradiation) is manifested by thickening of the epidermis, especially the stratum corneum, which disrupts the western differentiation of keratin [26,27]. Furthermore, changes in transepidermal water loss are observed to be an acute stress response under the action of external aging factors such as ultraviolet radiation [28]. In addition to specific changes in skin appearance and function, the lipid processing of keratinocytes declines with age. This causes quantitative and qualitative changes in lipophilic molecules within the SC membrane [29].

The skin plays an important role as a barrier between the harsh external environment and the host. Certainly, the two barriers are essential for survival: the barrier to water and electrolytes (osmotic barrier) as well as the barrier to invading and toxic microorganisms (antibacterial barrier). Lipids play a crucial role in the formation and maintenance of these barriers [30]. The hydrophobic extracellular lipid matrix in the stratum corneum is mainly composed of Cers, cholesterol, and free FAs, providing a barrier to the movement of water and electrolytes [31]. A variety of lipids, such as FAs, monoglycerides, SPs, phospholipids, and especially FAs, have antibacterial activity and contribute to the formation of antibacterial barriers [32].

As a type of lipid on the surface of the skin, FAs is an important ingredient that maintains the function of the skin barrier. Symptoms of FAs deficiency include dryness of the epidermis, peeling, loose skin, skin inflammation, increased sensitivity to irritation, and slow healing [33]. A recent study of FAs in the skin epidermis found an age-dependent content of the major lipid components, with the main difference being the level of polyunsaturated fatty acids [34]. This is different from our finding that the differential fatty acids are saturated fatty acids such as stearic acid and palmitic acid. It is interesting to note that, unlike C15:0 and C17:0, which were shown in a study on aging to decrease in FAs on the skin’s surface [35], our study was screened to find that the age difference was mostly an even number of long-chain FAs. Previous studies have shown that free FAs present in sebum (such as lauric acid C12:0, palmitic acid C16:0, and oleic acid C18:0) can induce sebaceous cells to express antimicrobial peptides, affecting the inflammatory and antibacterial processes of the skin [36]. FAs derived from fish oil are thought to be involved in skin photoprotection [36]. Other mechanisms by which omega-3 PUFAs inhibit UV-induced keratinocyte damage may be the modulation of COX-2, NF-κb, and mitogen-activated protein kinase (MAPK)/extracellular signal-regulated kinase (ERK) pathways [37]. According to the results of this study, among females, the level of FAs on the surface of the right-hand skin was lower in the middle-aged group compared to the younger group. The overall condition of the skin—its surface texture, color, and physiological properties—is determined by factors such as hydration (i.e., a sufficient amount of water in the stratum corneum), sebum content, and surface acidity. Middle-aged women, especially in rural areas, are basically inseparable from housework; thus, inevitably, they undergo long-term exposure to various detergents. After detergent exposure, not only is the PH of the skin affected [38], but the FAs content of the skin also changes. Compared with those before exposure, after exposure, the long-chain FAs are reduced, the density of the lipid tissue is lower, and the layered structure of the extracellular stratum cornea is changed, resulting in skin barrier dysfunction [39,40].

Decreased levels of free FAs and triglycerides in the SC of aging skin are consistent with previous studies [41]. These skin lipid reductions may be caused by UV exposure affecting the activity of several cytokines. These cytokines include interleukin (IL)-6, IL-8, placental growth factor, and C-C base multiplex chemokines (CCL)2/MCP-3 produced by ultraviolet irradiation on keratinocytes or fibroblasts [42]. As previous studies have shown, the difference in lipid levels and lipatopoietic enzymes in SC fat in light-damaged forearm skin and light-protected buttock skin is not due to differences in anatomical location but rather due to the effects of ultraviolet radiation [43]. One study suggests that age-dependent adaptation of the upper glycolysis and glycerol metabolism interface has a potential effect on epidermal barrier function [44]. This decrease in glycerol metabolism can have serious consequences for the lipid layer of the cuticle epidermal barrier and may lead to the impaired barrier function of aging skin [45]. In one study simulating UV irradiation, the most significant lipid changes were found to be increases in long-chain Cers, LysoPC, and glycerides, while phospholipids such as phosphatidylcholine (PC) and phosphatidylethanolamine (PE) did not follow a clear increase/decrease pattern [46]. Similarly, one Japanese study found a trend toward decreased skin ceramide and cholesterol in older adults. The skin is the target organ for hormones [47]. Ovarian estrogen, which accounts for the majority of estrogen in women, declines with age, especially during menopause. Postmenopausal estrogen deficiency can lead to skin atrophy and accelerate skin aging [48]. Early studies have shown that sterols are closely related to skin conditions, such as acne [49,50]. Recent studies also suggested that the majority of differentiating lipid species were diacylglycerols (DGs), followed by fatty acyls, sterols, and prenols, which are more abundant in the skin of acne patients [51]. To our knowledge, no studies have explored the association between age and sterols and PRs [52].

A previous study using time-of-flight secondary ion mass spectrometry to examine age-related changes in human stratum corneum lipids found significant age-related increases and changes in the spatial distribution of sterol-cholesterol sulfate, a membrane-stable lipid [25]. However, in our study, no differences in this substance were found between different age groups. Our results suggest that sphingoid bases may suggest a novel age-related metabolic pathway. Sphingomyelins are a class of lipids that contain long-chain (C16-C20) amino alcohols (sphingosine bases). As the backbone structure of all SPs (i.e., schingolipids and phospholipids), Cers are metabolites including sphingosine bases, sphingosine-1-phosphate, and Cers-1-phosphate [53]. Sphingosine regulates cellular function by sphingosine kinase (sphk1 and sphk2) 1-phosphate (S1P), which is synthesized by sphingosine and by S1P receptor-dependent and non-dependent pathways. It has been found that sphingosine groups can be used in conjunction with other lipids as biomarkers for FAs to predict atopic dermatitis in children [54]. Meanwhile, a large number of studies have shown that the pharmacological regulation of Cers and sphingosines in the skin helps to treat skin diseases associated with the abnormal proliferation and differentiation of the epidermis and inflammation [55].

Aging is a complex and variable process, which is difficult to quantify and interpret. This study used lipidomics to explore the mechanism of skin barrier function damage with age, quantify the degree of damage, and find important related lipid metabolism, which provides a theoretical basis for the development and optimization of corresponding anti-aging hand skincare products. Moreover, the study of aging from the perspective of lipids provides a scientific basis for the development of geriatric dermatology. This will enable us to further guide clinical medication and the prevention of skin diseases. However, due to the limitations of observational studies, more in-depth mechanistic studies determining whether the observed effects on the metabolites reported are due to epigenetic regulation will be performed in our future studies.

## 5. Conclusions

There were differences in the right-hand SSL of females at different ages, and the differences may be related to the metabolism of GLs and sphingoid bases.

## Figures and Tables

**Figure 1 metabolites-13-00553-f001:**
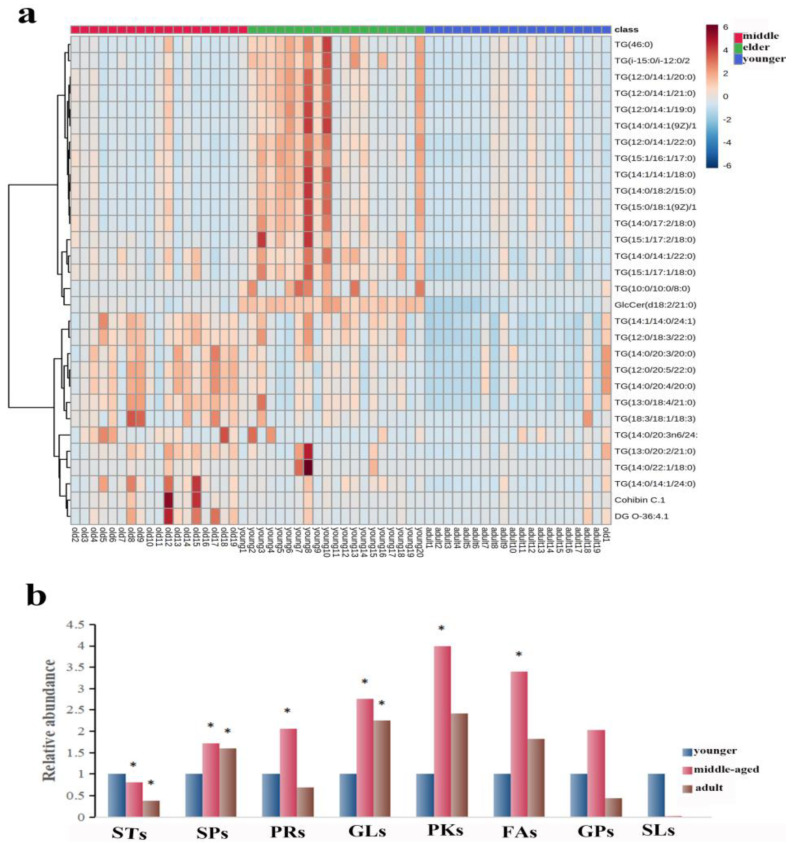
(**a**): Heatmap illustrating the distribution of SSL among females at different ages. (**b**): The relative amounts of the eight major lipids in different age groups. The sign of * indicates that the difference is statistically significant when compared with the younger group (*p* < 0.05). Abbreviations: STs, sterol lipids; SPs, sphingolipids; PRs, prenol lipids; GLs, glycerolipids; PKs, polyketides; FAs, fatty acids; GPs, glycerophospholipids; SLs, saccharolipids.

**Figure 2 metabolites-13-00553-f002:**
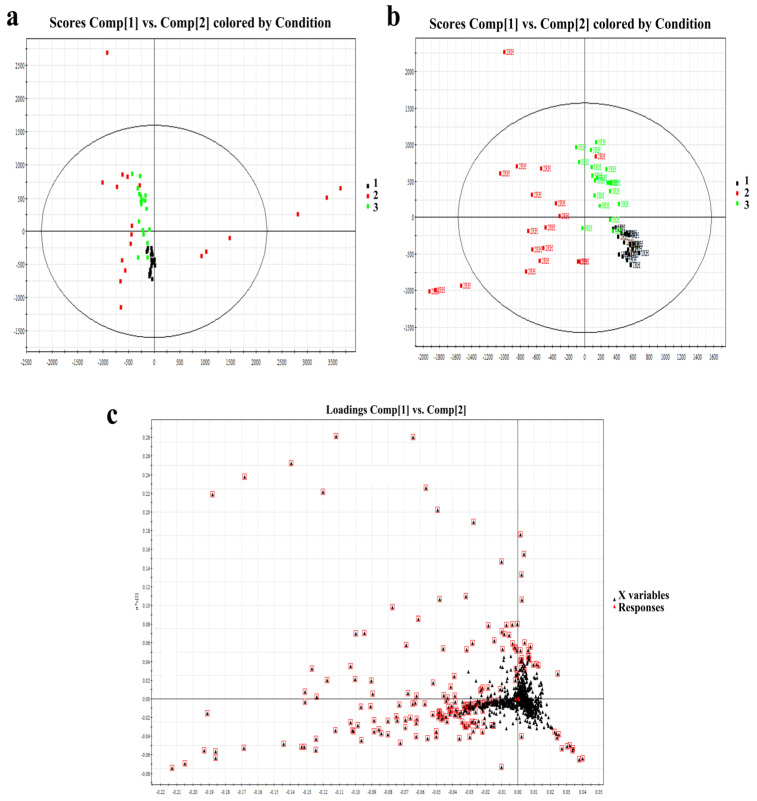
(**a**): PCA score of right-hand SSL among different-aged females. (**b**): PLS-DA score of right-hand SSL among different-aged females. RH means right hand. (**c**): Loading plot of selected components of right-hand SSL among different-aged females in ESI+. Data points marked by red boxes can be seen as the characteristic ions of female SSL in each age group, respectively. 1, younger group; 2, middle-aged group; 3, elder group.

**Figure 3 metabolites-13-00553-f003:**
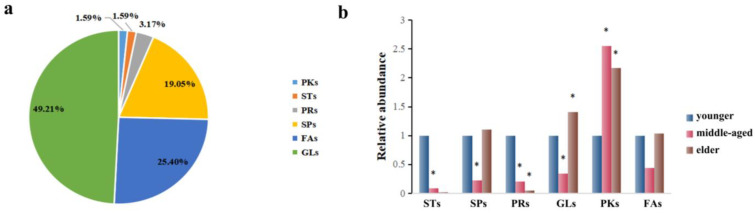
(**a**): The distribution of the important individual lipid species responsible for the discrimination of samples. (**b**) Comparison of relative abundance of differentiated metabolites of SSL in different age groups. The sign of * indicates that the difference is statistically significant when compared with the younger group (*p* < 0.05).

**Figure 4 metabolites-13-00553-f004:**
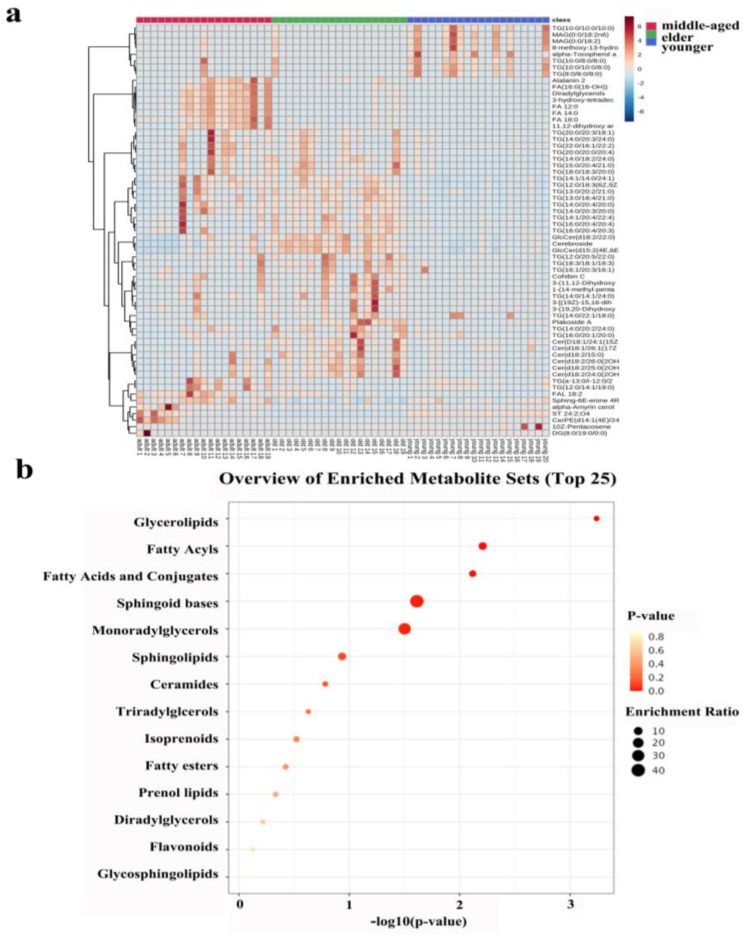
(**a**): Heatmap illustrating the distribution of SSL among females at different ages. (**b**): Metabolic pathway of important individual lipids among different age groups.

**Table 1 metabolites-13-00553-t001:** The gradient conditions of mobile phases.

Time (min)	A Phase (%)	B Phase (%)
0.0	80	20
1.0	80	20
5.0	40	60
11.0	20	80
18.0	10	90
19.0	0	100
21.0	0	100
21.1	80	20
22.0	80	20

**Table 2 metabolites-13-00553-t002:** The most important individual lipid species responsible for the discrimination between skin surface lipid samples collected from three groups.

Retention Time (min)	*m*/*z*	Formula	Description	Anova (*p*)	Highest Mean
0.8523	262.2384	C_14_H_28_O_3_	3-hydroxy-tetradecanoic acid	0.000	middle-aged
0.8523	218.2124	C_12_H_24_O_2_	FA(12:0)	0.000	middle-aged
1.8561	302.3059	C_18_H_36_O_2_	FA(18:0)	0.000	middle-aged
0.9692	290.2702	C_16_H_32_O_3_	FA(16:0(16-OH))	0.000	middle-aged
0.9692	246.2436	C_14_H_28_O_2_	FA(14:0)	0.000	middle-aged
4.3784	282.2800	C_18_H_32_O	FAL(18:2)	0.000	middle-aged
13.2548	575.5057	C_37_H_66_O_4_	3-(11,12-Dihydroxy-15,19-dotriacontadienyl)-5-methyl-2(5H)-furanone, 9ci	0.000	elder
13.7980	577.5213	C_37_H_68_O_4_	3-[(19Z)-15,16-dihydroxydotriacont-19-en-1-yl]-5-methyl-5H-furan-2-one	0.000	elder
6.6815	355.2852	C_21_H_38_O_4_	MAG(0:0/18:2)	0.000	younger
7.3001	355.2852	C_21_H_38_O_4_	MAG(0:0/18:2n6)	0.000	younger
5.6706	368.4260	C_25_H_50_	10Z-Pentacosene	0.002	younger
13.2617	577.5207	C_37_H_68_O_4_	Cohibin C	0.002	elder
5.8905	327.2537	C_19_H_34_O_4_	FA(19:2;O2)	0.006	middle-aged
13.7980	551.5058	C_35_H_66_O_4_	3-(19,20-Dihydroxytriacontyl)-5-methyl-2(5H)-furanone	0.006	elder
1.2578	274.2747	C_16_H_32_O_2_	Diradylglycerols	0.046	middle-aged
1.2509	362.3267	C_20_H_40_O_4_	11,12-dihydroxy arachidic acid	0.048	middle-aged
13.8186	876.8028	C_55_H_102_O_6_	TG(14:1/14:0/24:1)	0.000	middle-aged
13.2479	874.7869	C_55_H_100_O_6_	TG(12:0/18:3(6Z,9Z,12Z)/22:0)	0.000	elder
14.4376	904.8344	C_57_H_106_O_6_	TG(13:0/20:2/21:0)	0.000	elder
12.7256	872.7716	C_55_H_98_O_6_	TG(13:0/18:4/21:0)	0.000	elder
13.2548	900.8024	C_57_H_102_O_6_	TG(14:0/20:4/20:0)	0.000	elder
12.7325	881.7627	C_57_H_100_O_6_	TG(12:0/20:5/22:0)	0.000	elder
13.8049	603.5368	C_39_H_70_O_4_	DG(O-36:4)	0.000	elder
13.8255	902.8183	C_57_H_104_O_6_	TG(14:0/20:3/20:0)	0.000	middle-aged
7.9599	572.4903	C_33_H_62_O_6_	TG(10:0/10:0/10:0)	0.000	younger
14.4376	878.8193	C_55_H_104_O_6_	TG(14:0/14:1/24:0)	0.000	elder
7.3070	544.4583	C_31_H_58_O_6_	TG(10:0/10:0/8:0)	0.000	younger
14.4582	956.8639	C_61_H_110_O_6_	TG(20:0/20:3/18:1)	0.000	middle-aged
15.7642	960.8970	C_61_H_114_O_6_	TG(14:0/20:2/24:0)	0.000	elder
6.1036	488.3956	C_27_H_50_O_6_	TG(8:0/8:0/8:0)	0.001	younger
15.0217	958.8799	C_61_H_112_O_6_	TG(14:0/20:3/24:0)	0.001	middle-aged
6.6746	516.4270	C_29_H_54_O_6_	TG(10:0/8:0/8:0)	0.001	younger
15.7504	934.8814	C_59_H_112_O_6_	TG(16:0/20:1/20:0)	0.001	elder
12.2370	901.7284	C_59_H_96_O_6_	TG(14:1/20:4/22:4)	0.001	elder
12.7325	903.7431	C_59_H_98_O_6_	TG(16:0/20:4/20:4)	0.001	elder
13.2617	905.7588	C_59_H_100_O_6_	TG(16:0/20:4/20:3)	0.001	elder
15.0767	932.8662	C_59_H_110_O_6_	TG(14:0/18:2/24:0)	0.002	elder
11.8041	894.7567	C_57_H_96_O_6_	TG(18:3/18:1/18:3)	0.003	elder
15.0905	906.8505	C_57_H_108_O_6_	TG(14:0/22:1/18:0)	0.004	elder
13.9492	928.8351	C_59_H_106_O_6_	TG(15:0/20:4/21:0)	0.004	elder
14.4720	930.8490	C_59_H_108_O_6_	TG(18:0/18:3/20:0)	0.006	elder
12.7187	782.7254	C_48_H_92_O_6_	TG(a-13:0/i-12:0/20:0)	0.009	middle-aged
15.6336	986.9108	C_63_H_116_O_6_	TG(22:0/16:1/22:2)	0.009	middle-aged
15.0630	984.8953	C_63_H_114_O_6_	TG(20:0/20:0/20:4)	0.017	middle-aged
12.3264	780.7093	C_48_H_90_O_6_	TG(12:0/14:1/19:0)	0.020	middle-aged
12.2714	870.7570	C_55_H_96_O_6_	TG(16:1/20:3/16:1)	0.023	elder
6.6609	516.4614	C_30_H_58_O_5_	DG(8:0/19:0/0:0)	0.039	middle-aged
9.6592	832.2423	C_39_H_43_O_20_	Alatanin 2	0.023	middle-aged
8.0079	473.4000	C_31_H_52_O_3_	alpha-Tocopherol acetate	0.000	younger
13.1654	822.8074	C_56_H_100_O_2_	alpha-Amyrin cerotate	0.018	middle-aged
8.5101	838.7146	C_50_H_95_NO_8_	Cerebroside	0.000	elder
6.6540	740.6045	C_43_H_81_NO_8_	GlcCer(d15:2(4E,6E)/22:0)	0.000	elder
5.4162	413.2673	C_18_H_37_NO_6_S	Sphing-6E-enine 4R-sufate	0.000	middle-aged
7.3688	522.4893	C_33_H_63_NO_3_	Cer(d18:2/15:0)	0.000	elder
9.7967	678.6408	C_43_H_83_NO_4_	Cer(d18:2/25:0(2OH))	0.000	elder
7.4169	782.6516	C_46_H_87_NO_8_	GlcCer(d18:2/22:0)	0.000	elder
11.9829	948.7871	C_57_H_105_NO_9_	Plakoside A	0.000	elder
9.7760	648.6302	C_42_H_81_NO_3_	Cer(D18:1/24:1(15Z))	0.001	elder
9.4457	664.6251	C_42_H_81_NO_4_	Cer(d18:2/24:0(2OH))	0.001	elder
8.0972	750.6111	C_40_H_81_N_2_O_7_P	CerPE(d14:1(4E)/24:0(2OH))	0.002	middle-aged
10.4633	676.6615	C_44_H_85_NO_3_	Cer(d18:1/26:1(17Z))	0.003	elder
10.1333	692.6557	C_44_H_85_NO_4_	Cer(d18:2/26:0(2OH))	0.010	elder
21.6564	391.2851	C_24_H_38_O_4_	ST(24:2;O4)	0.000	younger

## Data Availability

The data presented in this study are available in article and Appendix A.

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
