# Peer review of "Lipid Differences and Related Metabolism Present on the Hand Skin Surface of Different-Aged Asiatic Females—An Untargeted Metabolomics Study"

_metabolites, 2023, doi:10.3390/metabo13040553_

Round 1

Reviewer 1 Report

The authors propose the use of modern tools for the study of differences in the amount and type of lipids in women at different ages, they present the paper Lipid differences and related metabolism present on the hand skin surface of different age females-an untargeted metabolomics study.

The proposal is interesting since they make use of new technologies for lipidomics and metabolomics. In this sense I have some concerns:

1) The design of the study. The authors mention the inclusion and exclusion criteria, which are well defined; however, in the discussion section they mention several factors that can alter the percentages and type of lipids that they present in their results. Among them are hormonal cycles, mainly estrogen, different lifestyles, exposure to UV radiation, among others, so the question arises as to whether their results are influenced by these factors and that they were not considered to make their population more homogeneous. by etsudio.
  2) It is not clear if their population is from the same region, or they come from different regions, that is, if they are all from urban or rural regions.
3) The discussion section lacks a greater comparison between its results and the results obtained in other investigations and other populations.
4) It is not very clear what would be the implication of its results in the context of skin health and the aging process.
5) They should improve their writing in English.
6) Some figure legends do not correspond to the image to which reference is made, as in figure 2.
7) It is not mentioned on which characteristics the age groups that were formed are based, only a reference is indicated, but I consider that it is necessary to explain it even a little.

Author Response

Q1: The design of the study. The authors mention the inclusion and exclusion criteria, which are well defined; however, in the discussion section they mention several factors that can alter the percentages and type of lipids that they present in their results. Among them are hormonal cycles, mainly estrogen, different lifestyles, exposure to UV radiation, among others, so the question arises as to whether their results are influenced by these factors and that they were not considered to make their population more homogeneous. by etsudio.

A1: 

Thank you very much for your review. In fact, the difference in skin lipids can be affected by many factors. In the course of life, these factors are basically unavoidable. For hormones, levels were not consistent across age groups, so we show grouped results. For UV exposure, although we selected subjects with less than four hours of outdoor activity during the study design phase, the accumulation of UV damage with age is worth considering. 

Q2:  It is not clear if their population is from the same region, or they come from different regions, that is, if they are all from urban or rural regions.

A2: Thank you for your suggestions. All the subjects were from rural areas. We have added it to the revised manuscript.

Q3: The discussion section lacks a greater comparison between its results and the results obtained in other investigations and other populations.

A3: Thank you for your constructive suggestions. We have added some comparisons with the results of others to the discussion section of the revised manuscript. Hopefully, it will make our manuscript better.

Q4: It is not very clear what would be the implication of its results in the context of skin health and the aging process.

A4: 

Thank you for your review. One of the most common skin and beauty problems is skin aging. Most studies on skin aging have focused on changes in physical parameters such as transepithelial water loss, surface pH, or skin hydration. However, the literature on age-related changes in SC lipid composition is scarce and the results are inconsistent. The in-depth study of aging from the perspective of lipids, using the results of population studies, provides theoretical support for skin health and aging. The results of this study show that cosmetics or medical personnel can provide specific lipid supplementation according to age, and the metabolic pathways involved provide ideas for the development of skin care or therapeutic products.

Q5: They should improve their writing in English.

A5: Thank you for your suggestions. Sorry for the inappropriate language writing. We strove to improve the language in the revised manuscript.

Q6: Some figure legends do not correspond to the image to which reference is made, as in figure 2.

A6: Sorry for the unclear description. We have revised the manuscript in the hope of making it clearer.

Q7:  It is not mentioned on which characteristics the age groups that were formed are based, only a reference is indicated, but I consider that it is necessary to explain it even a little.

A7: Thank you for your suggestions. According to your suggestion, we have added relevant content to the revised manuscript. As follows: “Due to age differences in the biophysical characteristics of skin, in this study, according to the guidance of WHO and the actual situation in our country, the subjects were divided into three age groups to analyze the changes of SSL.”

The attachment is the revised manuscript for your review, which hopefully meets the publication requirements. Thanks again for your review.

Reviewer 2 Report

This manuscript reported the differences in skin surface lipid (SSL) of different age females by lipidomics.

The authors are expected to consider following points before publication: 

1. Why hand skin was studied? why not other skin area? Hand skin might be quite different from other skin area in terms of stratum corneum thickness or skin microbiota. Authors are expected to discuss this in the introductin or discussion section.  

2. It is mentioned in the introduction that this study utilized non-invasive skin detection technology and lipidomics to access the skin barrier function of people of different ages. However, there is no related data on skin barrier function. Thus, authors are expected to characterized the skin barrier of  volunteers; or to discuss in details on the correlation between SSL and skin barrier function. 

Author Response

Q1: Why hand skin was studied? why not other skin area? Hand skin might be quite different from other skin area in terms of stratum corneum thickness or skin microbiota. Authors are expected to discuss this in the introductin or discussion section.  

A1: Thank you very much for your constructive suggestions.In the revised manuscript, we have added a discussion on the selection of skin regions in the introduction section. As follows: “Due to the richest bacterial diversity, less restricted in membership, and susceptibility to temporal variability, the hand skin is different from other regions of the skin and was explored as a sampling site in this study.”

Q2:  It is mentioned in the introduction that this study utilized non-invasive skin detection technology and lipidomics to access the skin barrier function of people of different ages. However, there is no related data on skin barrier function. Thus, authors are expected to characterized the skin barrier of  volunteers; or to discuss in details on the correlation between SSL and skin barrier function. 

A2: Sorry for the lack of data on the skin barrier. Following your suggestion, we have added a detailed discussion of the correlation between the skin barrier and SSL to the discussion section of the revised manuscript. We will increase the collection of these data in our subsequent research in order to provide better support for research in the skin field. Thank you very much for your constructive suggestions. As follows: “The skin plays an important function of providing a barrier between the harsh external environment and the host. The two absolutely necessary for survival are the barrier to water and electrolyte movement (osmotic barrier) and the barrier to invading and toxic microorganisms (antibacterial barrier). Lipids play a crucial role in the formation and maintenance of these barriers [28]. The hydrophobic extracellular lipid matrix in the stratum corneum is mainly composed of ceramides, cholesterol, and free fatty acids, providing a barrier to the movement of water and electrolytes [29]. A variety of lipids, such as fatty alcohols, monoglycerides, sphingolipids, phospholipids, and especially free fatty acids, have antibacterial activity and contribute to the formation of antibacterial barriers [30].”

Thanks again for your review and constructive suggestions. 

Reviewer 3 Report

The manuscript entitled “Lipid differences and related metabolism present on the hand skin surface of different age females- an untargeted metabolomics study” by Tian Chen et al., was presented as original article in which the authors, adopting a cross-sectional design, performed a skin-related lipidomic analysis to explore the metabolic pathways on skin barrier function. Although the study design is appropriate, a significant writing quality improvement is required for this manuscript. The authors have to revise the manuscript more thoroughly, especially the abstract part, and make sure to revise all other sections including the conclusion part. The article needs to be rewritten focusing the broad profiling experiment. Some of the experimental details are repeated both in the methods and results sections.

The study is mainly observational and lacks an in-depth mechanistic insight that further validates the use of Mass Spectrometry (MS) in conducting such analysis. It would be appropriate performing mechanistic studies to determine whether the observed effects on the metabolites reported are due to epigenetic regulation.

Material and Methods section: specify the code, commercial brand, city, and country of provenance for all materials.

The figures are very low-quality. Characters in the boxes are not legible in real size.

Please, double check and correct the typos of words/acronyms/abbreviations in the text. The reviewer suggests: I) to capitalize the first letters of full words that define the acronyms/abbreviations; II) to introduce the acronyms for the terms that are repeated two or more times in the text. Once you have entered an acronym, make sure that you no longer repeat the extended form in the text. 

Author Response

Q1: Although the study design is appropriate, a significant writing quality improvement is required for this manuscript. The authors have to revise the manuscript more thoroughly, especially the abstract part, and make sure to revise all other sections including the conclusion part. The article needs to be rewritten focusing the broad profiling experiment. Some of the experimental details are repeated both in the methods and results sections.

A1: Sorry for the inappropriate description. We have done our effort to review and revise the content and syntax. Hopefully it will be easy to read. For the repeated parts of the methods and results, improvements have been made in the revised manuscript.

Q2: The study is mainly observational and lacks an in-depth mechanistic insight that further validates the use of Mass Spectrometry (MS) in conducting such analysis. It would be appropriate performing mechanistic studies to determine whether the observed effects on the metabolites reported are due to epigenetic regulation.

A2: Thank you very much for your constructive suggestions. In fact, what you said is indeed the limitation of observational research, and the specific mechanism still needs to be further explored. We will consider the role of epigenetic regulation in this in our future mechanism research according to your suggestion. And we added this as a limitation at the end of the discussion of the revised manuscript.

Q3: Material and Methods section: specify the code, commercial brand, city, and country of provenance for all materials.

A3: Thank you for your advice. We have improved it according to your request in the methods section of the revised manuscript. Sorry the characters in the picture are not clear. We have made new displays in the revised manuscript in the hope of making it easier to read.

Q4: The figures are very low-quality. Characters in the boxes are not legible in real size.

A4: Thank you for your advice. We have readjusted the font and arrangement in the figures in the revised manuscript, hoping to make it easy to read.

Q5: Please, double check and correct the typos of words / acronyms / abbreviations in the text. 

A5: Sorry for the inappropriate use of acronyms. We have made improvements in the revised manuscript that we hope will be presented correctly.

Reviewer 4 Report

This is a very interesting scientific work that aims to know how skin composition can modify during different ages. These data could be promoted new specific treatments per age. However, the number of participants is too low to speak of the variability of the expression of some lipids. Furthermore, the population is Asiatic; these can be race-related data: specify Asiatic females in the work’s title. Given these brief considerations, the work looks well performed, although doesn’t provide scientific support to dermatological pathologies, it can be the starting point to understanding how our body varies during various age groups.

line 41-45, you should add some references, such as: doi: 10.1111/exd.14276.

The work is, in my opinion ready to be published after minor revisions.

Author Response

Q: This is a very interesting scientific work that aims to know how skin composition can modify during different ages. These data could be promoted new specific treatments per age. However, the number of participants is too low to speak of the variability of the expression of some lipids. Furthermore, the population is Asiatic; these can be race-related data: specify Asiatic females in the work’s title. Given these brief considerations, the work looks well performed, although doesn’t provide scientific support to dermatological pathologies, it can be the starting point to understanding how our body varies during various age groups.

line 41-45, you should add some references, such as: doi: 10.1111/exd.14276.

The work is, in my opinion ready to be published after minor revisions.

A: Thank you for your suggestions. We changed the title in the revised manuscript to "Lipid differences and related metabolism present on the hand skin surface of different age Asiatic females- an untargeted metabolomics study ". And the references you suggested have been added to the revised manuscript. In addition, we have made an effort to check and improve the language of the manuscript. It is hoped that the revised manuscript will meet the requirements for publication. 

Round 2

Reviewer 1 Report

The authors made the suggested corrections, and the changes in the text are consistent with the corrections made.
Authors must make some minor spell corrections.

Author Response

Thanks again for your review and constructive suggestions. As much as possible, we revised and improved the writing. Hopefully, our changes will be satisfactory to you.

Reviewer 3 Report

The authors solved most of my concerns, so the reviewer approves the manuscript for publication.

Author Response

(The authors gave the same response as above.)
